# One-Pot Synthesis of 1-Thia-4-azaspiro[4.4/5]alkan-3-ones via Schiff Base: Design, Synthesis, and Apoptotic Antiproliferative Properties of Dual EGFR/BRAF^V600E^ Inhibitors

**DOI:** 10.3390/ph16030467

**Published:** 2023-03-22

**Authors:** Lamya H. Al-Wahaibi, Essmat M. El-Sheref, Mohamed M. Hammouda, Bahaa G. M. Youssif

**Affiliations:** 1Department of Chemistry, College of Sciences, Princess Nourah Bint Abdulrahman University, Riyadh 11564, Saudi Arabia; 2Chemistry Department, Faculty of Science, Minia University, El Minia 61519, Egypt; 3Department of Chemistry, College of Science and Humanities in Al-Kharj, Prince Sattam Bin Abdulaziz University, Al-Kharj 11942, Saudi Arabia; 4Chemistry Department, Faculty of Science, Mansoura University, Mansoura 35516, Egypt; 5Pharmaceutical Organic Chemistry Department, Faculty of Pharmacy, Assiut University, Assiut 71526, Egypt

**Keywords:** thioglycolic acid, schiff base, spiro, apoptosis, antiproliferative, mechanism

## Abstract

In this investigation, novel 4-((quinolin-4-yl)amino)-thia-azaspiro[4.4/5]alkan-3-ones were synthesized via interactions between 4-(2-cyclodenehydrazinyl)quinolin-2(1*H*)-one and thioglycolic acid catalyzed by thioglycolic acid. We prepared a new family of spiro-thiazolidinone derivatives in a one-step reaction with excellent yields (67–79%). The various NMR, mass spectra, and elemental analyses verified the structures of all the newly obtained compounds. The antiproliferative effects of **6a**–**e**, **7a**, and **7b** against four cancer cells were investigated. The most effective antiproliferative compounds were **6b**, **6e**, and **7b**. Compounds **6b** and **7b** inhibited EGFR with IC_50_ values of 84 and 78 nM, respectively. Additionally, **6b** and **7b** were the most effective inhibitors of BRAF^V600E^ (IC_50_ = 108 and 96 nM, respectively) and cancer cell proliferation (GI_50_ = 35 and 32 nM against four cancer cell lines, respectively). Finally, the apoptosis assay results revealed that compounds **6b** and **7b** had dual EGFR/BRAF^V600E^ inhibitory properties and showed promising antiproliferative and apoptotic activity.

## 1. Introduction

The development of new drugs in anti-cancer research depends on a better understanding of druggable targets. According to this strategy, changing particular cancer biomarkers will produce beneficial therapeutic effects [1]. Selective anti-cancer medications must be more effective at destroying tumors while having fewer side effects on normal cells [2]. Single-target therapy has been recognized as causing chemotherapeutic resistance [3]. Recently, combination therapy (drug cocktails combining two or more therapeutic agents with different modes of action and non-overlapping toxicities) was authorized as an alternative to single-target chemotherapy for cancer [4]. Despite the possibility of additive and synergistic effects, combination therapy frequently results in unexpected adverse effects, such as increased toxicity. Alternatives to combination therapy include drugs with two or more targets, a reduced risk of drug interactions, enhanced pharmacokinetics (PK), and improved safety profiles [5]. Additionally, a dual-target kinase may inhibit drug interactions, harmful off-target impacts, poor patient compliance, and elevated costs of production [5].

Combining tyrosine kinase (TK) and BRAF inhibitors has proven useful in preventing tumor growth and minimizes resistance in clinical trials. Combining vemurafenib and EGFR inhibitors in thyroid carcinoma may help overcome resistance to BRAF inhibitors [6]. This combination has also been effective in BRAF^V600E^ colorectal cancer [7]. Additionally, several in vitro substances have been developed, including EGFR/VEGFR-2 and BRAF, which contain the essential pharmacophoric groups necessary to suppress tyrosine kinase [8,9]. In summary, dual/multi-targeted kinase inhibitors can be turned into effective anti-cancer medications [10].

In medicinal chemistry, nitrogen heterocycles are crucial structural components. The chemical and biological uses of quinoline, one of many heterocyclic compounds, have been studied by numerous research groups [11,12]. The FDA recently approved quinazoline derivatives, such as gefitinib and erlotinib (Figure 1), as EGFR inhibitors for treating non-small cell breast and lung cancers [13,14,15]. In addition, Thr766 in the EGFR pocket forms a water-mediated hydrogen bond with the nitrogen atom at position 3 of the quinazoline core [16,17]. New quinoline derivatives, such as pelitinib and neratinib, which are potent EGFR inhibitors, were developed through the bioisosteric replacement of the quinazoline core with a quinoline ring (Figure 1) [18,19,20,21,22]. The bioisosteric substitution did not require water molecules to facilitate the interaction with amino acid residue Thr766 [23].

We previously reported synthesizing two novel series of quinoline-2-one-based derivatives as potential apoptotic antiproliferative agents targeting the EGFR inhibitory pathway [24]. Compounds **I** and **II** (Figure 1) inhibited EGFR effectively, with IC_50_ values of 0.18 and 0.09 µM, respectively. Furthermore, the two compounds significantly increased the apoptotic markers caspase-3, caspase-8, and the Bax levels while decreasing antiapoptotic Bcl2. In another study [25], compound **III**, a quinoline-based derivative, was developed as a dual EGFR and BRAF^V600E^ inhibitor with IC_50_ values of 1.30 and 3.8 µM, respectively.

The Spiro scaffold is another building material with potential applications in medicinal chemistry. Chemists have introduced a ring for rigidity when designing new drugs to potentially reduce the entropic penalties upon binding to target proteins. Conformational restriction can also be achieved by spiro ring fusion. They are frequently used in the design and discovery of drugs due to their inherent three-dimensionality and novel structural characteristics. Previously, spiro ring systems were efficiently integrated into enzyme inhibitors, with kinases topping the list, as well as protein–protein interaction inhibitors [26,27,28].

On the other hand, the thiazole moiety has piqued researchers’ interest due to its robust biological properties [29,30]. Regarding anti-cancer activity, thiazole and its derivatives rank among the most active compounds [31,32,33,34,35,36]. Furthermore, several clinically available anti-cancer drugs contain thiazole-containing compounds, such as dabrafenib (**IV**, Figure 2) (BRAF inhibitor) [37].

Abdel-Maksoud et al. also developed a novel series of thiazole-based derivatives for testing as BRAF^V600E^ inhibitors. Compound **V** (Figure 2) exhibited the most potent antiproliferative activity, with potent BRAF^V600E^ inhibitory activity and an IC_50_ value of 0.05 µM [38]. We recently reported the synthesis of novel thiazole-based derivatives that act as dual EGFR and BRAF^V600E^ inhibitors [39]. Compound **VI** (Figure 2) demonstrated significant antiproliferative activity against EGFR and BRAF^V600E^, with IC_50_ values of 74 and 107 nM, respectively.

Following our previous study on dual targeting strategies and motivated by the findings above [40,41,42], we synthesized new spiro-compounds that combine quinolinone, thiazolidinone, and spiro-cyclic in a single molecule via the reaction of thioglycolic acid and 4-(2-cyclodenehydrazinyl)4uinoline-2(1*H*)-one. As a result, two new compounds, **6a**–**e** (Scaffold A) and **7a** and **7b** (Scaffold B), were developed to generate new potent antiproliferative agents targeting EGFR and/or BRAF^V600E^, as shown in Figure 3.

The new compounds underwent a cell viability assay test to determine their influence on the vitality of normal cell lines. They were also tested against a panel of four cancer cell lines as antiproliferative agents. Furthermore, the most active compounds were tested for antiproliferative activity against EGFR and BRAF^V600E^. Finally, we tested the most active compounds for apoptotic potential against caspase 3, caspase 8, Bax, and the antiapoptotic Bcl2.

## 2. Results and Discussion

### 2.1. Chemistry

We aimed to develop a novel series of spiro-compounds, namely **6a**–**e**, **7a** and **7b**. Figure 1 depicts the steps taken to obtain our target 4-((6-substituted-2-oxo-1,2-dihydroquinolin-4-yl)amino)-1-thia-4-azaspiroalkan-3-ones (**6a**–**e**) in high yields via the interaction between 4-(2-cyclodenehydrazinyl)5uinoline-2(1*H*)-ones (**5a**–**e**) [43,44,45] and thioglycolic acid refluxed under dry benzene with a free catalyst at a molar ratio of 1:20 for 20 h.

**Reagents and reaction conditions**: (**a**) POCl_3_/stirring for 1 h, 70 °C; (**b**) AcOH/H_2_O/refluxing overnight; (**c**) Hydrazine/EtOH/refluxing for 3 h; (**d**) Cyclic ketone/EtOH/refluxing for 3 h; I Thioglycolic acid/Benzene/refluxing for 24 h.

The structures of our obtained products, **6a–e**, were confirmed using NMR, mass spectrometry, and elemental analysis. All of the spectral data show that the acquired molecular formula for **6a–e** comprises one molecule from compounds **5a**–**e** and one molecule of thioglycolic acid, with the elimination of the H_2_O molecule. Compound **6b** [4-((2-oxo-1,2-dihydroquinolin-4-yl)amino)-1-thia-4-azaspiro-[4.5]decan-3-one], with the chemical formula C_17_H_19_N_3_O_2_S and a detectable molecular ion peak at *m/z* = 329, was used as a representative example. Moreover, the ^1^H NMR spectra of compound **6b** revealed two broad singlet signals at δ_H_ = 11.01 and 10.07 ppm, respectively, indicative of NH-1 and exo-NH-4b. Two singlet signals at δ_H_ = 6.11 and 3.74 ppm were assigned as H-3 and H-2′, respectively, in addition to quinolinone-CH between 7.97 and 7.12 ppm (m, 4H). The ^13^C NMR spectra for compound **6b** had four downfield-lying lines at δ_C_ = 167.56 (C-3′), 162.85 (C-2), 151.76 (C-4), and 139.06 (C-8a) ppm, respectively. C-5 resonated at δ_C_ = 61.27 ppm and was characteristic of a spiro-structure.

The possibility of an isomeric structure in compound **6b′** (Figure 4) was ruled out by ^1^H NMR, which revealed an NH chemical shift at 10.07 ppm due to hydrogen bonding with the carbonyl group (C-3′). Furthermore, the ^15^N NMR spectrum revealed a signal at δ_N_ = 131.7 ppm, which produced an HSQC correlation with the proton at δ_H_ = 10.07 ppm. An HMBC correlation with two protons at δ_H_ = 10.07 and 6.11 ppm, assigned as NH-4b and H-3, was also indicated (Figure 5). The latter applies to structure **6b** but not **6b′**. Furthermore, the two signals at δ_N_ = 142.1 ppm, designated as NH-1, show an HSQC correlation with the proton at H 6.11 ppm, designated as H-3. These correlations are not possible for structure **6b′**.

We also investigated the reaction of thioglycolic acid with 4-(2-(2,2,6,6-tetramethylpiperidin-4-ylidene)hydrazinyl)quinolin-2(1*H*)-one (**5f**) and 4-(2-(1-benzylpiperidin-4-ylidene) hydrazine-yl)quinolin-2(1*H*)-one (**5g**). The reaction followed the same pattern and produced similar products, **7a** and **7b**, in high yields (Figure 2).

For example, consider compound **7b**, which was designated as 8-benzyl-4-((2-oxo-1,2-dihydroquinolin-4-yl)amino)-1-thia-4,8-diazaspiro[4.5]decan-3-one with the molecular formula C_23_H_24_N_4_O_2_S (Figure 6, Table 1). The ^1^H NMR of compound **7b** revealed that H-3 was distinctive as the vinylic singlet at δ_H_ 6.08 ppm, with its attached carbon appearing at δ_C_ 92.33 ppm. Furthermore, H-3 exhibits an HMBC correlation with both protonated nitrogens at δ_N_ 141.9 and 132.0 ppm. The latter could be N-1 and N-4b, although it is unclear which was which. Furthermore, N-1 exhibits an HMBC correlation with the proton resonance at δ_H_ 7.28 ppm, implying that H-8 is present. Of the two carbons that exhibit an HSQC correlation with this proton, the one at δ_C_ 115.45 ppm exhibits an HMBC correlation with the protons in the quinoline ring, designated as C-8. The only visible 1H doublet in the aromatic region appears at δ_H_ 7.98 ppm and is assigned to H-5, whose attached carbon appears at δ_C_ 122.44 ppm. H-5 establishes a COSY correlation with the dd at δ_H_ 7.12 ppm, designated as H-6. The attached carbon appears at δ_C_ 120.31 ppm in H-6, establishing a COSY correlation with the other dd, at δ_H_ 7.46, designated as H-7. Attached carbon appears at δ_C_ 130.13 ppm in H-7. In turn, H-7 establishes a COSY correlation with H-8. The carbon at δ_C_ 139.23 establishes an HMBC correlation with H-5 and H-7, designated as C-8a. The carbon at δ_C_ 112.26 establishes a strong HMBC correlation with H-3, NH-4b, H-6, and H-8, designated as C-4a. These six HMBC correlations are three-bond correlations. The carbon at δ_C_ 162.90 establishes an HMBC correlation with H-3 and (weakly) NH-4b, designated as C-2. The carbon at δ_C_ 149.51 establishes an HMBC correlation with H-3, NH-4b, H-5, and (weakly) H-8, designated as C-4. This arrangement constituted the quinolinone structure.

Furthermore, H-8″ is distinctive as the 2H singlet at δ_H_ 3.56, whose attached carbon appears at δ_C_ 61.36 ppm., and establishes an HMBC correlation with the signal at δ_C_ 128.73 ppm, designated as C-*o*. Both C-*o* and C-*m* establish HSQC correlations with the m-signal at δ_H_ 7.35 ppm. Furthermore, the carbon at δ_C_ 126.93 establishes a COSY correlation with both H-*o* and H-*m*, and HSQC correlations with the signal at 7.28 ppm. This carbon must be C-*p*, as shown in Table 1.

Compounds **6a**–**e** were synthesized by cyclizing the corresponding hydrazones **5a**–**e** with thioglycolic acid according to the suggested mechanism in Figure 3. Compounds **5a**–**c** were catalyzed with thioglycolic acid to form tertiary carbocation 8 and a nucleophilic addition of HS-thioglycolic, followed by rearrangement with the loss of an H+-proton to produce the intermediate 9. Nucleophilic attacks from NH-2 to the carbonyl group eliminated an H_2_O molecule, yielding the corresponding products 6a–e via intermediates 9 and 10. Compounds **7a** and **7b** were also formed via a similar mechanism.

### 2.2. Biology

#### 2.2.1. Cell Viability Assay

The epithelial (MCF-10A) cell line of a normal human mammary gland was used to test the viability of the new substances. Compounds **6a**–**e**, **7a**, and **7b** were incubated on MCF-10A cells for four days before being tested for viability using the MTT assay [46,47]. According to Table 2, none of the tested substances revealed cytotoxic impacts, and the cell viability for the compounds tested at 50 µM was >89%.

#### 2.2.2. Antiproliferative Assay

The antiproliferative role of **6a**–**e**, **7a**, and **7b** was tested against the four human cancer cell lines, namely MCF-7 (breast cancer cell line), Panc-1 (pancreatic cancer cell line), A-549 (lung cancer cell line), and HT-29 (colon cancer cell line), using the MTT assay and erlotinib as the reference drug [48,49]. Table 2 shows the median inhibitory concentration of each tested compound (IC_50_).

Overall, the recently screened derivatives **6a**–**e**, **7a**, and **7b** showed encouraging antiproliferative action against the four cancer cell lines tested, with an average IC_50_ (GI_50_) ranging between 32 and 81 nM compared to the reference erlotinib, which had a GI_50_ of 33 nM. Compounds **6b**, **6e**, and **7b** had the highest antiproliferative activity, with GI_50_ values of 35, 40, and 32 nM, respectively. Compound 7b (R = Ph-CH_2_, R^1^ = R^2^ = H, Scaffold B) was the most effective derivative, with a GI_50_ value of 32 nM. It was comparable to, and even more effective than, erlotinib (GI_50_ = 33 nM) against the MCF-7 cell line, as shown in Table 3. The other Scaffold B compound 7a (R = H, R^1^ = R^2^= CH_3_) was the least effective derivative, with GI_50_ values of 81 nM. It was 2.5-fold less potent than **7b**, suggesting the significance of the N-benzyl piperidine moiety for antiproliferative activity.

Compounds **6b** (R = H, *n* = 1, Scaffold A) and 6e (R = OCH_3_, *n* = 1, Scaffold A) were the second and third most active compounds, with GI_50_ values of 35 and 40 nM, respectively. They were 1.1- and 1.25-fold less potent than **7b**. Additionally, compound **6c** (R = CH_3_, *n* = 1, Scaffold A) showed modest antiproliferative activity with a GI_50_ value of 73 nM against the four cancer cell lines tested. Compound 6c was 2.1- and 1.8-folds less potent than 6b and 6e, respectively.

Compound **6a** (R = H, *n* = 0, Scaffold A) was 1.3-fold less potent than compound **6b** (R = H, *n* = 1, Scaffold A) against the four cancer cell lines, with a GI50 value of 45 nM. The same pattern was observed when comparing compounds 6d (R = OCH3, *n* = 0, Scaffold A) and 6e (R = OCH3, *n* = 1, Scaffold A). Compound **6d** had a GI50 of 52 nM, whereas 6e had a GI50 of 40 nM. These results showed that the spiro moiety’s ring size and the type of substitution at the quinoline moiety’s 6-position significantly impacted the antiproliferative function. Regarding the ring size, the cyclohexyl ring moiety was more tolerated for antiproliferative activity than the cyclopentyl ring moiety. Furthermore, the nature of substitution at the quinoline 6-position was associated with an increase in the antiproliferative activity, in the order H > OCH3 > CH3.

#### 2.2.3. EGFR Inhibitory Assay

As potential targets for their antiproliferative activity, compounds **6a**–**e**, **7a**, and **7b** were tested for EGFR inhibitory activity [50]. Table 3 displays these results as IC_50_ values. The outcomes of this test align with the antiproliferative test, where the two most effective antiproliferative derivatives, compounds **7b** (R = Ph-CH_2_, R^1^ = R^2^ = H, Scaffold B) and **6b** (R = H, *n* = 1, Scaffold A), were the most effective EGFR inhibitors, with IC values of 78 ± 05 and 84 ± 06 nM, respectively. They were also equipotent to the reference erlotinib (IC_50_ = 80 ± 05).

Compounds **6a** (R = H, *n* = 0, Scaffold A), **6d** (R = OCH_3_, *n* = 0, Scaffold A), and **6e** (R = OCH_3_, *n* = 1, Scaffold A) displayed significant anti-EGFR activity with IC_50_ values of 97 ± 07, 104 ± 08, and 92 ± 07 nM, respectively. They were 1.2- to 1.3-folds less potent than erlotinib.

Once again, compounds **6c** (R = CH_3_, *n* = 1, Scaffold A) and **7a** (R = H, R^1^ = R^2^ = CH_3_, Scaffold B) had the lowest activity as EGFR inhibitors, with IC_50_ values of 135 ± 11 and 149 ± 12 nM, respectively. These findings demonstrated that compounds **6b** and **7b** are workable antiproliferative candidates with substantial EGFR inhibitory properties.

#### 2.2.4. BRAF^V600E^ Inhibitory Assay

The anti-BRAF^V600E^ activity of compounds **6a**–**e**, **7a**, and **7b** was also investigated in vitro [51], using erlotinib as a reference compound; the results are displayed in Table 3. According to the enzyme testing results, the investigated compounds had moderate to good BRAF^V600E^ inhibitory activity, with IC_50_ values between 96 and 187 nM. In every instance, the compounds were less effective than the standard drug erlotinib (IC_50_ = 60 nM).

Table 3 shows that the most potent EGFR inhibitors, compounds **7b** (R = Ph-CH_2_, R^1^ = R^2^ = H, Scaffold B) and **6b** (R = H, *n* = 1, Scaffold A), were also the most potent BRAF^V600E^ inhibitors, with IC values of 96 ± 8 and 108 ± 9 nM, respectively. Compounds **7b** and **6b** were roughly equivalent to erlotinib as antiproliferative agents; however, as BRAF^V600E^ inhibitors, they were 1.6- and 1.8-fold less potent. Compounds **6a**, **6c**, **6d**, **6e**, and **7a** showed weak to moderate anti-BRAF activity with IC_50_ values ranging between 129 and 187 nM, as shown in Table 3. The in vitro assay findings revealed that compounds 6b and 7b are potent antiproliferative agents that can act as dual EGFR/BRAF^V600E^ inhibitors.

#### 2.2.5. Apoptotic Markers Activation Assay

Numerous biochemical and morphological processes are involved in apoptosis, also called programmed cell death [52]. Antiapoptotic proteins such as Bcl-2 and Bc-W coexist with proapoptotic proteins such as Bad and Bax [53]. Proapoptotic proteins stimulate the release of cytochrome-c, whereas antiapoptotic proteins control apoptosis by inhibiting the release of cytochrome-c. The outer mitochondrial membrane becomes permeable when the ratio of proapoptotic proteins exceeds that of antiapoptotic proteins, setting off a series of events. The release of cytochrome c triggers both caspase-3 and caspase-9. Then, caspase-3 induces apoptosis by attacking several of the vital proteins that the cell requires [53].

##### Caspase 3 Activation Assay

Compounds **6b** and **7b** were tested as caspase-3 activators against the human pancreatic (Panc-1) cancer cell line [54]; the findings are shown in Table 4. Our results showed that derivatives **6b** and **7b** had a significant overexpression of caspase-3 protein levels (487.50 ± 4 and 544.50 ± 5 pg/mL, respectively) compared to the reference staurosporine (503.00 ± 4 pg/mL). The highest active substance, **7b**, caused caspase-3 protein overexpression (544.50 ± 5 pg/mL) in the Panc-1 cancer cell line, which was 8.5-fold higher than the control with untreated cells, and even higher than staurosporine. Compound **6b** (487.50 ± 4 pg/mL) showed comparable caspase-3 activation to the reference staurosporine. The above findings suggest that apoptosis may have contributed to the antiproliferative action of the tested compounds and caspase-3 overexpression.

##### Caspase-8, Bax and Bcl-2 Levels Assay

Compounds **6b** and **7b** were investigated for their impact on the Bacl-2, Bax, and caspase-8 levels against the Panc-1 cancer cell line, with staurosporine as a reference [55]. Our results are presented in Table 5.

Our findings demonstrated that the investigated substances significantly increased the Bax and caspase-8 levels compared to staurosporine. Compound **7b** had the highest level of caspase-8 overexpression (1.90 ng/mL), followed by compound **6b** (1.50 ng/mL) and the reference staurosporine (1.80 ng/mL). Additionally, **7b** showed a comparable induction of Bax (295 pg/mL) to staurosporine (280 pg/mL), which was 37-fold higher than the control with untreated Panc-1 cancer cells. Finally, compound 7b caused the equipotent down-regulation of the Bcl-2 protein level (1.00 ng/mL), followed by compound 6b (1.30 ng/mL) in the Panc-1 cell line, and the reference staurosporine (1.10 ng/mL). The apoptosis assay results revealed that compounds **6b** and **7b** possess dual EGFR/BRAF inhibitory properties and show promising antiproliferative and apoptotic activity.

## 3. Conclusions

We designed and synthesized a small set of novel compounds, 4-((quinolin-4-yl)amino)thia-azaspiro[4.4/5]alkan-3-ones **6a**–**e**, **7a**, and **7b**, in order to develop new dual-targeting antiproliferative agents. The MTT assay was used to investigate the antiproliferative effects of the new compounds against a panel of four cancer cell lines. Compounds **6b**, **6e**, and **7b** had higher antiproliferative activity, with GI_50_ values of 35, 40, and 32 nM, than erlotinib, which had a GI_50_ of 33 nM. The most potent compounds were then tested for EGFR and BRAF^V600E^ inhibition. Our results showed that compounds **6b**, **6e**, and **7b** were the most potent EGFR inhibitors, with IC_50_ values of 84, 92, and 78 nM, respectively. Furthermore, **6b**, **6e**, and **7b** showed promising BRAF^V600E^ inhibitory activity, with IC_50_ values of 108, 129, and 96 nM, respectively. Therefore, they can be considered effective antiproliferative agents that act as dual EGFR/BRAF^V600E^ inhibitors. The most active derivative in Panc-1 cells, **7b**, induces apoptosis via caspase 3 overexpression and an increased ratio of *Bax/Bcl-2* genes compared to the control cells. In the future, more in vitro and in vivo studies, as well as chemical modifications, may be required to develop highly effective antiproliferative agents.

## 4. Experimental

### 4.1. Chemistry

General details: See Appendix A.

The 4-(2-substitutedhydrazinyl)-6-substituted-quinolin-2(1*H*)-one **5a**–**g** were prepared as reported [36,37,38], the thioglycolic acid (Aldrich) was used as received.

General procedure for preparation of compounds **6a**–**e** and **7a**,**b**

The **5a**–**g** (1 mmol) and thioglycolic acid (20 mmol) were refluxed in 20 mL dray benzene in a round-bottom flask for 20 h. The reaction mixture was allowed to cool, and a white precipitate formed. The precipitate was suction filtrated and washed four times with 25 mL of 3% Na_2_CO_3_ solution to remove unreacted thioglycolic acid before being thoroughly dried to yield the products **6a**–**e**, **7a**, and **7b**.

#### 4.1.1. 4-((2-Oxo-1,2-dihydroquinolin-4-yl)amino)-1-thia-4-azaspiro[4.4]nonan-3-one (**6a**)

White ppt (69%), m.p 215–17 °C; ^1^H NMR (DMSO-d_6_): δ_H_ = 11.01 (bs, 1H; NH-1), 10.07 (bs, 1H; NH-4b), 7.97 (d, *J* = 8.1 Hz; 1H, H-5), 7.47 (dd, *J* = 8.1, 7.2 Hz; 1H, H-7), 7.28 (d, *J* = 7.8 Hz; 1H, H-8), 7.13 (t, *J* = 8, 7.2 Hz, H-6), 6.09 (s, 1H; H-3), 3.74 (s, 2H; H-2′), 1.1–1.95 ppm (m, 8H; cyclopentyl-CH_2_), ^13^C NMR (DMSO-d_6_): δ_C_ = 167.54 (C-3′), 162.89 (C-2), 149.69 (C-4), 139.05 (C-8a), 130.64 (C-7), 122.09 (C-5), 120.93 (C-6), 115.54 (C-8), 111.98 (C-4a), 92.64 (C-3), 54.40 (C-2′), 47.12 (C-5′), 34.11, 26.11, 25.25 ppm (Cyclopentyl-CH_2_), ^15^N NMR (DMSO-d_6_): δ_N_ = 142 (N-1), 128,6 (N-4b), 93.3 ppm (N-4′). *m/z*, Ms = 315 (M^+^, 12). *Anl. Calcd. For* C_16_H_17_N_3_O_2_S: C, 60.93; H, 5.43; N, 13.32; S, 10.17. Found: C, 61.08; H, 5.33; N, 13.43; S, 10.29.

#### 4.1.2. 4-((2-Oxo-1,2-dihydroquinolin-4-yl)amino)-1-thia-4-azaspiro[4.5]decan-3-one (**6b**)

White ppt (70%), m.p 194–96 °C; ^1^H NMR (DMSO-d_6_): δ_H_ = 11.05 (bs, 1H; NH-1), 10.09 (bs, 1H; NH-4b), 7.97 (m, 1H; H-5), 7.49 (m, 1H; H-6), 7.27 (m, 1H; H-7), 7.12 (m, 1H; H-8), 6.11 (s, 1H; H-3), 3.74 (s, 2H; H-2′), 1.11–1.92 ppm (m, 10H; cyclohexyl-CH_2_), ^13^C NMR (DMSO-d_6_): δ_C_ = 167.56 (C-3′), 162.85 (C-2), 149.76 (C-4), 139.06 (C-8a), 130.34 (C-7), 122.09 (C-5), 120.69 (C-6), 115.53 (C-8), 111.99 (C-4a), 92.66 (C-3), 54.42 (C-2′), 47.17 (C-5′), 34.18, 26.17 ppm (Cyclohexyl-CH), ^15^N NMR (DMSO-d_6_): δ_N_ = 142.1 (N-1), 131,7 (N-4b), 93.5 ppm (N-4′). *m/z*, Ms = 329 (M^+^, 10). *Anl. Calcd.* For C_17_H_19_N_3_O_2_S: C, 61.98; H, 5.81; N, 12.76; S, 9.73. Found: C, 62.11; H, 5.77; N, 12.83; S, 9.66.

#### 4.1.3. 4-((6-Methyl-2-oxo-1,2-dihydroquinolin-4-yl)amino)-1-thia-4-azaspiro[4.5]decan-3-one (**6c**)

White ppt (75%), m.p 160–62 °C; ^1^H NMR (DMSO-d_6_): δ_H_ = 10.93 (bs, 1H; NH-1), 10.05 (bs, 1H; NH-4b), 7.78(s, 1H, H-8), 7.33 (m, 1H; H-8), 7.17 (m, 1H; H-5), 6.07 (s, 1H; H-3), 3.24 (s, 2H; H-2′), 1.11–1.93 (m, 10H; Ali-CH), 2.11 ppm (s, 3H, CH_3_), ^13^C NMR (DMSO-d_6_): δ_C_ = 167.49 (C-3′), 162.15 (C-2), 149.75 (C-4), 139.16 (C-8a), 131.48 (C-7), 129.60 (C-6), 121.77 (C-5), 115.45 (C-8), 111.84 (C-4a), 92.61 (C-3), 54.33 (C-2′), 46.98 (C-5′), 33.90, 25.98 (Cyclohexyl-CH), 20.58 ppm (CH_3_). *m/z*, Ms = 343 (M^+^, 10). *Anl. Calcd. For* C_18_H_21_N_3_O_2_S: C, 62.95; H, 6.16; N, 12.23; S, 9.34. Found: C, 63.10; H, 6.22; N, 12.43; S, 9.21.

#### 4.1.4. 4-((6-Methoxy-2-oxo-1,2-dihydroquinolin-4-yl)amino)-1-thia-4-azaspiro[4.4]nonan-3-one (**6d**)

White ppt (67%), m.p 236–38 °C; ^1^H NMR (DMSO-d_6_): δ_H_ = 11.12 (bs, 1H; NH-1), 10.07 (bs, 1H; NH-4b), 7.99–7.11 (m, 3H; H-5,7,8), 6.06 (s, 1H; H-3), 3.70 (s, 3H; OMe), 3.35 (s, 2H; H-2′), 1.12–1.95 ppm (m, 8H; Cyclpentyl-CH_2_), ^13^C NMR (DMSO-d_6_): δ_C_ = 167.52 (C-3′), 162.84 (C-2), 149.11 (C-4), 136.07 (C-8a), 133.40 (C-6), 128.28 (C-7), 119.52 (C-5), 115.52 (C-8), 111.98 (C-4a), 92.67 (C-3), 55.55 (C-2′), 54.94 (OMe), 47.44 (C-5′), 32.77, 25.90, 23.77 ppm (Cyclopentyl-CH). *Anl. Calcd. For* C_17_H_19_N_3_O_3_S: C, 59.11; H, 5.54; N, 12.17; S, 9.28. Found: C, 59.23; H, 5.44; N, 11.99; S, 9.18.

#### 4.1.5. 4-((6-Methoxy-2-oxo-1,2-dihydroquinolin-4-yl)amino)-1-thia-4-azaspiro[4.5]decan-3-one (**6e**)

White ppt (79%), m.p 238–40 °C; ^1^H NMR (DMSO-d_6_): δ_H_ = 10.93 (bs, 1H; NH-1), 10.07 (bs, 1H; NH-4b), 7.50–7.34 (m, 1H; H-,8), 7.22–7.13 (m, 2H; H-5,7), 6.06 (s, 1H; H-3), 3.71 (s, 3H; OMe), 3.33 (s, 2H; H-2′), 1.14–1.92 ppm (m, 10H; Cyclohexyl-CH_2_), ^13^C NMR (DMSO-d_6_): δ_C_ = 167.60 (C-3′), 162.49 (C-2), 149.49 (C-4), 136.90 (C-8a), 133.46 (C-6), 128.28 (C-7), 119.52 (C-5), 116.82 (C-8), 112.29 (C-4a), 92.99 (C-3), 55.62 (C-2′), 55.42 (OMe), 47.56 (C-5′), 33.0, 25.94, 23.91 ppm (Cyclohexyl-CH). *Anl. Calcd. For* C_18_H_21_N_3_O_3_S: C, 60.15; H, 5.89; N, 11.69; S, 8.92. Found: C, 60.25; H, 5.77; N, 11.80; S, 8.88.

#### 4.1.6. 7,7,9,9-Tetramethyl-4-((2-oxo-1,2-dihydroquinolin-4-yl)amino)-1-thia-4,8-diazaspiro-[4.5]decan-3-one (**7a**)

White ppt (78%), m.p 165–67 °C; ^1^H NMR (DMSO-d_6_): δ_H_ = 11.07 (bs, 1H; NH-1), 10.26 (s, 1H; NH-4b), 7.78 (t, 1H; H-5), 7.65 (m,1H; H-7), 7.05 (m, 2H; NH, H-6), 6.05 (s, 1H; H-3), 3.51 (s, 2H; H-2′), 2.67 (s, 4; H-6′), 2.44 (s, 12H; 4CH_3_), ^13^C NMR (DMSO-d_6_): δ_C_ = 167.57 (C-3′), 162.68 (C-2), 149.68 (C-4), 139.02 (C-8a), 130.35 (C-7), 122.14 (C-5), 120.72 (C-6), 115.54 (C-8), 111.99 (C-4a), 92.87 (C-3), 55.55 (C-2′), 47.11 (C-5′),34.0 (C-6′), 26.33, 25.25 ppm (CH_3_). *Anl. Calcd. For* C_20_H_26_N_4_O_2_S: C, 62.15; H, 6.78; N, 14.50; S, 8.30. Found: C, 62.29; H, 6.88; N, 14.38; S, 8.36.

#### 4.1.7. 8-Benzyl-4-((2-oxo-1,2-dihydroquinolin-4-yl)amino)-1-thia-4,8-diazaspiro[4.5]decan-3-one (**7b**)

White ppt (71%), m.p 180–82 °C; ^1^H NMR (DMSO-d_6_): δ_H_ = 11.01 (bs; 1H, NH-1), 10.09 (s; 1H, NH-4b), 7.98 (d, *J* = 8.1 Hz; 1H, H-5), 7.46 (dd, *J* = 7.7, 7.5 Hz; 1H, H-7), 7.35 (m; 4H, H-*o*, *m*), 7.28 (m; 3H, H-8, *p*), 7.12 (dd, *J* = 7.7, 7.4 Hz; 1H, H-6), 6.08 (s; 1H, H-3), 3.71 (s; 2H, H-2′), 3.56 (s; 2H, H-8″), 2.67 (t, *J* = 5.6 Hz; 2H, H-6′/6″), 2.55 (m; 4H, H-7′/7″), 2.44 ppm (t, *J* =5.4 Hz; 2H, H-6″/6′), ^13^C NMR (DMSO-d_6_): δ_C_ = 167.33 (C-3′), 162.90 (C-2), 149.51 (C-4), 139.23 (C-8a), 138.31 (C-*i*), 130.13 (C-7), 128.73 (C-*o*), 128.16 (C-*m*), 126.93 (C-*p*), 122.44 (C-5), 120.31 (C-6), 115.45 (C-8), 112.26 (C-4a), 92.33 (C-3), 61.36 (C-8″), 54.11 (C-2′), 53.23, 52.07 (C-7′/7″), 47.33 (C-5′), 34.46 (C-6′/6″), 27.26 ppm (C-6″/6′). *Anl. Calcd. For* C_23_H_24_N_4_O_2_S: C, 65.69; H, 5.75; N, 13.32; S, 7.62. Found: C, 65.51; H, 5.89; N, 13.22; S, 7.55.

### 4.2. Biology

#### 4.2.1. Cell Viability Assay

The normal human mammary gland epithelial (MCF-10A) cell line was used to test the viability of the new compounds [46,47]. See Appendix A.

#### 4.2.2. Antiproliferative Assay

The antiproliferative activity of **6a**–**e**, **7a**, and **7b** was tested against the four human cancer cell lines, Panc-1 (pancreatic cancer cell line), MCF-7 (breast cancer cell line), HT-29 (colon cancer cell line), and A-549 (lung cancer cell line), using the MTT assay and Erlotinib as the reference drug [48,49]. See Appendix A.

#### 4.2.3. EGFR Inhibitory Assay

Compounds **6a–e**, **7a**, and **7b** were tested for EGFR inhibitory activity as a potential target for their antiproliferative activity [50]. See Appendix A.

#### 4.2.4. BRAF^V600E^ Inhibitory Assay

The anti-BRAF^V600E^ activity of compounds **6a–e**, **7a**, and **7b** was also investigated in vitro [51] using Erlotinib as a reference compound. See Appendix A.

#### 4.2.5. Apoptotic Markers Assay

##### Caspase 3 Activation Assay

Compounds **6b** and **7b** were tested as caspase-3 activators against the human pancreatic (Panc-1) cancer cell line [52,53,54]. See Appendix A.

##### Caspase-8, Bax and Bcl-2 Levels Assay

Compounds **6b** and **7b** were further investigated for their impact on the caspase-8, Bax, and Bacl-2 levels against the Panc-1 cancer cell line and staurosporine as a reference [55]. See Appendix A.

## Data Availability

The data will be provided upon request.

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
