# Peer review of "One-Pot Synthesis of 1-Thia-4-azaspiro[4.4/5]alkan-3-ones via Schiff Base: Design, Synthesis, and Apoptotic Antiproliferative Properties of Dual EGFR/BRAFV600E Inhibitors"

_pharmaceuticals, 2023, doi:10.3390/ph16030467_

Round 1

Reviewer 1 Report (Previous Reviewer 2)

I would like to extend my appreciation to Lamya et al. for refining their manuscript on the synthesis of novel and potentially effective EGFR/BRAFV600E inhibitors.  While most of the issues has been fixed, several concerns are still remaining:

(1) Kindly review all scheme/figure files carefully, as some of them appear blurry or contain zigzags. Additionally, Figures 4 and 5 differ in style from the others, and the font sizes of the text vary too much from one scheme to another. Please make the necessary adjustments to achieve uniformity.

(2) It is not advisable to include a handwritten figure in the main text. Please edit it accordingly.

(3) Some of the integration of the NMR spectra lacks labeling/normalization. Please rectify this.

Author Response

Reviewer 1

Comments and Suggestions for Authors

I would like to extend my appreciation to Lamya et al. for refining their manuscript on the synthesis of novel and potentially effective EGFR/BRAFV600E inhibitors.  While most of the issues has been fixed, several concerns are still remaining:

1- Kindly review all scheme/figure files carefully, as some of them appear blurry or contain zigzags. Additionally, Figures 4 and 5 differ in style from the others, and the font sizes of the text vary too much from one scheme to another. Please make the necessary adjustments to achieve uniformity.

Response:

We'd like to express our gratitude. All figures and schemes have been revised and appear to be in good condition; however, your version may have a problem; in any case, we will work with the production system to achieve better results.

(2) It is not advisable to include a handwritten figure in the main text. Please edit it accordingly.

Response:

Done as advised.

(3) Some of the integration of the NMR spectra lacks labeling/normalization. Please rectify this.

Response:

All NMR spectra were revised, and any necessary corrections were made.

Reviewer 2 Report (Previous Reviewer 3)

The present version is ok and can definitely be accepted after the language check

Author Response

Reviewer 2

Comments and Suggestions for Authors

The present version is ok and can definitely be accepted after the language check

Response:

We appreciate your considering our work for publication. We already upload the English editing certificate through MDPI system.

This manuscript is a resubmission of an earlier submission. The following is a list of the peer review reports and author responses from that submission.

Round 1

Reviewer 1 Report

The manuscript reports the synthesis of 7 compounds combining quinolinone, thiazolidinone, and spirocyclic in one molecule, their antiproliferative activities against four human cancer cell lines, and inhibition of dual-target kinases, EGFR and BRAFE. The topic is reasonably innovative, the methodology is scientific, the discussion is logical, and the conclusions are well supported by the experimental results. However, frankly saying, more structure variety of compounds is required to get a solid SAR conclusion. The manuscript is presented in good shape except for many grammar mistakes. The following suggestions are for the authors to consider.

More recent references should be cited, such as the followings, but not limited to, up to the authors’ decision:

1.      Recent in vivo advances of spirocyclic scaffolds for drug discovery, Expert Opin Drug Discov. 2022, 17(6), 603-618.

2.      Spirocyclic Scaffolds in Medicinal Chemistry, J. Med. Chem. 2021, 64(1), 150–183

3.      Thiazole Ring—A Biologically Active Scaffold, Molecules. 2021,26(11), 3166.

4.      Synthesis and biological evaluation of new derivatives of thieno-thiazole and dihydrothiazolo-thiazole scaffolds integrated with a pyrazoline nucleus as anticancer and multi-targeting kinase inhibitor, RSC Adv., 2022, 12, 561-577

5.      Quinoline-based thiazolidinone derivatives as potent cytotoxic and apoptosis-inducing agents through EGFR inhibition, Chem Biol Drug Des, 2022, 99(4), 547-560

6.      Discovery of novel conjugates of quinoline and thiazolidinone urea as potential anti-colorectal cancer agent, J Enzyme Inhib Med Chem. 2022, 37(1), 2334-2347.

7.      A review on advances in synthetic methodology and biological profile of spirothiazolidin-4-ones, J Heterocyclic Chem. 2022, 59, 1839–1878.

In Scheme 1. Synthesis of azaspiroalkan-3-one 6a-e. I don’t think the steps from 1 to 5 are necessary if 1-5 are known compounds, and because in the Experimental (4.1), it was described that 5a-e were used as received.

The manuscript needs careful proofreading. I spotted some of the grammar mistakes but didn’t change all of them. (I might not be 100% right.) In addition, formats such as anticancer/anti-cancer, antiapoptotic/anti-apoptotic, hrs/h, Erlotinib /erlotinib, and American/ British English should be in the same style throughout the whole manuscript.

Table 1. needs reformatting to move some letters back to the end of the up line.

Table 4. Caspase-3 induction of compounds 6a and 6d. Here 6a and 6d should be 6b and 7b.

Why does Figure 5. 1H -15N HMBC for compound 6b. the same figure exit in both the main body and the supplementary?

Author Response

Reviewer #1

Comments and Suggestions for Authors

The manuscript reports the synthesis of 7 compounds combining quinolinone, thiazolidinone, and spirocyclic in one molecule, their antiproliferative activities against four human cancer cell lines, and inhibition of dual-target kinases, EGFR and BRAF. The topic is reasonably innovative, the methodology is scientific, the discussion is logical, and the conclusions are well supported by the experimental results. However, frankly saying, more structure variety of compounds is required to get a solid SAR conclusion. The manuscript is presented in good shape except for many grammar mistakes. The following suggestions are for the authors to consider.

More recent references should be cited, such as the followings, but not limited to, up to the authors’ decision:

  1. Recent in vivo advances of spirocyclic scaffolds for drug discovery, Expert Opin Drug Discov. 2022, 17(6), 603-618.
  2. Spirocyclic Scaffolds in Medicinal Chemistry, J. Med. Chem.2021, 64(1), 150–183
  3. Thiazole Ring—A Biologically Active Scaffold, Molecules. 2021,26(11), 3166.
  4. Synthesis and biological evaluation of new derivatives of thieno-thiazole and dihydrothiazolo-thiazole scaffolds integrated with a pyrazoline nucleus as anticancer and multi-targeting kinase inhibitor, RSC Adv.,2022, 12, 561-577
  5. Quinoline-based thiazolidinone derivatives as potent cytotoxic and apoptosis-inducing agents through EGFR inhibition, Chem Biol Drug Des, 2022, 99(4), 547-560
  6. Discovery of novel conjugates of quinoline and thiazolidinone urea as potential anti-colorectal cancer agent, J Enzyme Inhib Med Chem. 2022, 37(1), 2334-2347.
  7. A review on advances in synthetic methodology and biological profile of spirothiazolidin-4-ones, J Heterocyclic Chem. 2022, 59, 1839–1878.

Response:

Done as advised. The suggested references have been added to the revised version of the manuscript.

In Scheme 1. Synthesis of azaspiroalkan-3-one 6a-e. I don’t think the steps from 1 to 5 are necessary if 1-5 are known compounds, and because in the Experimental (4.1), it was described that 5a-e were used as received.

Response:

We'd like to thank the respected reviewer for this comment, which came after extensive revision. However, in order to provide the reader with a complete picture of the new compound synthesis, we only mention the complete scheme for the synthesis, with no discussion of synthesis of 1-5 compounds.

The manuscript needs careful proofreading. I spotted some of the grammar mistakes but didn’t change all of them. (I might not be 100% right.) In addition, formats such as anticancer/anti-cancer, antiapoptotic/anti-apoptotic, hrs/h, Erlotinib /erlotinib, and American/ British English should be in the same style throughout the whole manuscript.

Response:

The whole manuscript was revised, and all necessary corrections have been made.

Table 1. needs reformatting to move some letters back to the end of the up line.

Response:

Done as advised

Table 4. Caspase-3 induction of compounds 6a and 6d. Here 6a and 6d should be 6b and 7b.

Response:

Done as advised

Why does Figure 5. 1H -15N HMBC for compound 6b. the same figure exit in both the main body and the supplementary?

Response:

We used it as an example and to increase the reader's visibility.

Author Response

Reviewer #2

Comments and Suggestions for Authors

Thanks for Lamya et al. for their work on the synthesis of new promising EGFR/BRAFV600E inhibitors. The background is well introduced and relevant for the new design and following work. Most of the results are consistent yet there are limitations that warrant attention:

  1. Please apply the same style/fonts to all ChemDraw files including figures, synthetic schemes and especially the structures attached on NMR spectra.

Response:

Done as advised.

  1. Please keep the same size ratio for all figures/schemes. For example, Scheme 2 is larger than others.

Response:

Done as advised.

  1. Could you specify if all the reaction was running under air atmosphere or inert gas?

Response:

As demonstrated in the experimental section, all of our reactions occur under atmospheric pressure.

  1. The 1H NMR peaks in the aromatic region should be reported as individual peaks. For example, four “m” peaks for compound 6a instead of “m, 4H”.

Response:

Some samples underwent modification, but others did not display the pikes in a clear zoom and appeared as multiplies.

  1. Please show and normalize the integration on all 1H NMR. For example, no normalization: 6e, 7a, and etc. No integration: 6c, 6d, and etc. 6. Since mass spectroscopy data is available, could you consider reporting them as text in the experimental section? Also consider adjusting the display range of the mass data.

Response:

Indeed, in the supplementary data file, the integrations for all peaks are clearly displayed. In the 1H NMR, all values are present that include the full number of protons for each peak. However, in order to ensure the behavior of the reactions at the beginning of the experiment, we chose several samples (6a, 6b, and 6c) to be subjected to mass spectrometry. These results were further confirmed by the spectral data and elemental analyses. 

Reviewer 3 Report

The present paper describes the synthesis and activity evaluation of a small library of 1-thia-4-azaspiro[4.4/5]alkan-3-ones. Both parts are described in detail and appear internally coherent as well as properly supported by experimental data.

However, there are a few minor and major revisions to be made:

MINOR REVISIONS:

- In the title: “1-thia-4-azaspiro[4.4/5]alkan-3-ones” instead of “1-thia-4-azaspiro[4.4/5]alkan-3-one”

- In the affiliations: “to whom correspondence should be addressed” instead of “whom correspondence should be addressed”

- Pg. 1, in the keywords: “Schiff” instead of “schiff” (the same in the whole manuscript)

- Pg. 1, 1ST paragraph of the Introduction: a brief explanation of what combination therapy is should be included to allow the reader to better understand the paragraph

- Pg. 2, figure 1: the structure of Neratinib misses an oxygen on the 3-chloroaniline ring

- Pg. 2, line 15: a brief explanation of the role of the water molecules should be provided to justify the last sentence (The water molecule was not necessary in this bioisosteric substitution to facilitate interaction with the amino acid residue Thr766 [23]”)

- Pg. 3, line 6: “Spiro” instead of “Spiros”

- Pg. 3, line 7: “the last years” instead of “latest days”

- Pg. 3, Figure 2: the t-butyl moiety is missing from the structure of Dabrafenib

- Pg. 4, Figure 3: “X” should be substituted with “(CH2)n”, with n=0, 1. This should be made throughout the whole document

- Pg. 4, line 9: “aims at developing” instead of “goals to develop”

- Pg. 5, Scheme 1: an extra line is needed after the legend

- Pg. 5, line 4-7: the meaning of the sentence “On the other hand, we use compound 6b … and is detectable by mass spectrometry at m/z = 329” is not clear. It should be rephrased.

- The names of APIs should always be either capitalized or not throughout the whole documents, for consistency reasons

- The name BRAF or B-RAF should be used in the whole document, for consistency reasons

- Whenever molecules are referred to by their specific name, their numbers should be included in brackets (for instance, pg. 4 line 11: 4-((6-substituted-2-oxo-1,2-dihydroquinolin-4-yl)amino)-1-thia-4-azaspiroalkan-3-one (6a-e) instead of 4-((6-substituted-2-oxo-1,2-dihydroquinolin-4-yl)amino)-1-thia-4-azaspiroalkan-3-one 6a-e). In addition, the numbers of molecules should always be written in bold, especially in the Experimental Part where they are not clearly distinguishable from other numbers.

- In the Supplementary Material, the image titles should be written on the same page of the image they refer to (some are shifted in the previous one)

On the whole, the language is not always optimal. Many sentences, especially in the Results and discussion part where NMR spectra are referred to (pg. 5-9), are difficult to understand, also because some steps are omitted from the reasoning. From a theoretical viewpoint the conclusions are correct, but the choice of language seriously hampers the reader’s ability to understand the meaning.

In addition, plurals/singulars are often mismatched and verbs are not always correctly conjugated (for instance, the verb is missing from the first sentence of the Abstract.

Therefore, extensive rephrasing and elaboration is needed.

MAJOR REVISIONS:

- Pg. 10, second paragraph of 2.2.2: the sentence “even more effective than erlotinib against the MCF-7 cell line, as shown in Table 3” is an overstatement, because 78 and 80 nM are not strikingly different values

- Pg. 10, fourth paragraph of 2.2.2: while it is true that the ring size and substitution of the spiro ring affect the activity of the tested compounds, it is extremely arguable that there is a “significant” influence. Indeed, the IC50 values of the spiro compounds are of the same order of magnitude and their error intervals are very close

Author Response

Reviewer #3

Comments and Suggestions for Authors

The present paper describes the synthesis and activity evaluation of a small library of 1-thia-4-azaspiro[4.4/5]alkan-3-ones. Both parts are described in detail and appear internally coherent as well as properly supported by experimental data.

However, there are a few minor and major revisions to be made:

 MINOR REVISIONS:

1- In the title: “1-thia-4-azaspiro[4.4/5]alkan-3-ones” instead of “1-thia-4-azaspiro[4.4/5]alkan-3-one”

Response:

Done as advised.

2- In the affiliations: “to whom correspondence should be addressed” instead of “whom correspondence should be addressed”.

Response:

Done as advised.

3- Pg. 1, in the keywords: “Schiff” instead of “schiff” (the same in the whole manuscript)

Response:

Done as advised.

4- Pg. 1, 1ST paragraph of the Introduction: a brief explanation of what combination therapy should be included to allow the reader to better understand the paragraph

Response:

Done as advised.

5- Pg. 2, figure 1: the structure of Neratinib misses an oxygen on the 3-chloroaniline ring

Response:

Done as advised.

6- Pg. 2, line 15: a brief explanation of the role of the water molecules should be provided to justify the last sentence (The water molecule was not necessary in this bioisosteric substitution to facilitate interaction with the amino acid residue Thr766 [23]”)

Response:

The role of the water molecules already stated in the Introduction, Paragraph 3. (Highlighted in green in the revised manuscript)

7- Pg. 3, line 6: “Spiro” instead of “Spiros”

Response:

Done as advised.

8- Pg. 3, line 7: “the last years” instead of “latest days”

Response:

Done as advised.

9- Pg. 3, Figure 2: the t-butyl moiety is missing from the structure of Dabrafenib

Response:

Done as advised.

10- Pg. 4, Figure 3: “X” should be substituted with “(CH2)n”, with n=0, 1. This should be made throughout the whole document

Response:

Done as advised.

11- Pg. 4, line 9: “aims at developing” instead of “goals to develop”

Response:

Done as advised.

12- Pg. 5, Scheme 1: an extra line is needed after the legend

Response:

Done as advised.

13- Pg. 5, line 4-7: the meaning of the sentence “On the other hand, we use compound 6b … and is detectable by mass spectrometry at m/z = 329” is not clear. It should be rephrased.

Response:

Done as advised.

14- The names of APIs should always be either capitalized or not throughout the whole documents, for consistency reasons

Response:

Done as advised.

15- The name BRAF or B-RAF should be used in the whole document, for consistency reasons

Response:

Done as advised.

16- Whenever molecules are referred to by their specific name, their numbers should be included in brackets (for instance, pg. 4 line 11: 4-((6-substituted-2-oxo-1,2-dihydroquinolin-4-yl)amino)-1-thia-4-azaspiroalkan-3-one (6a-e) instead of 4-((6-substituted-2-oxo-1,2-dihydroquinolin-4-yl)amino)-1-thia-4-azaspiroalkan-3-one 6a-e). In addition, the numbers of molecules should always be written in bold, especially in the Experimental Part where they are not clearly distinguishable from other numbers.

Response:

Done as advised.

17- In the Supplementary Material, the image titles should be written on the same page of the image they refer to (some are shifted in the previous one)

 Response:

Done as advised.

18-On the whole, the language is not always optimal. Many sentences, especially in the Results and discussion part where NMR spectra are referred to (pg. 5-9), are difficult to understand, also because some steps are omitted from the reasoning. From a theoretical viewpoint the conclusions are correct, but the choice of language seriously hampers the reader’s ability to understand the meaning.

In addition, plurals/singulars are often mismatched, and verbs are not always correctly conjugated (for instance, the verb is missing from the first sentence of the Abstract.

Therefore, extensive rephrasing and elaboration is needed.

 Response:

The whole manuscript was revised, and we try our best to improve the language style.

 MAJOR REVISIONS:

- Pg. 10, second paragraph of 2.2.2: the sentence “even more effective than erlotinib against the MCF-7 cell line, as shown in Table 3” is an overstatement, because 78 and 80 nM are not strikingly different values

Response:

I'm sorry, sir, but 7b's IC50 value for MCF-7 is 34 nM and erlotinib's is 40 nM. However, when comparing 7b to erlotinib as an EGFR inhibitor, we noted that both are equipotent; both are underlined in grey.

- Pg. 10, fourth paragraph of 2.2.2: while it is true that the ring size and substitution of the spiro ring affect the activity of the tested compounds, it is extremely arguable that there is a “significant” influence. Indeed, the IC50 values of the spiro compounds are of the same order of magnitude and their error intervals are very close.

Response:

We appreciate the reviewer's input. Currently, we are preparing other derivatives for additional in vivo and in vitro studies as part of a future strategy, but the existing data may be regarded as primary data.

Round 2

Reviewer 1 Report

Many thanks to the authors for addressing the comments from the reviewers and providing a revised version.

1. The authors’ response claims, “The whole manuscript was revised, and all necessary corrections have been made.” However, there are still a lot of grammar mistakes in the main manuscript, supplementary and HIGHLIGHTS. I wonder if the authors read what I pointed out in the attached files last time. Some new errors are found, such as the cited reference numbers.

2. The authors added exactly the seven more references I suggested last time. I wish the authors had checked if they are relative and searched/added more. 

3. For Scheme 1, I am still confused if the authors prepared compounds 1 to 5 by themselves (following the literature methods) or, as it was described in the Experimental (4.1), that 5a-e were used as received. If 5a-e were purchased, how come the reaction steps from 1 to 5, including the conditions? It should be easy to clarify this.

4. As to Figure 5, to my understanding, generally, the supplementary presents what is not included in the main body. To save space on the pages and the readers’ time, one is OK. If the authors like to repeat it, I am happy with it. 

Author Response

Reviewer 1

Comments and Suggestions for Authors

Many thanks to the authors for addressing the comments from the reviewers and providing a revised version

  1. The authors’ response claims, “The whole manuscript was revised, and all necessary corrections have been made.” However, there are still a lot of grammar mistakes in the main manuscript, supplementary and HIGHLIGHTS. I wonder if the authors read what I pointed out in the attached files last time. Some new errors are found, such as the cited reference numbers.

Response:

Sir, we do our best to correct the grammatical errors either in the manuscript or in the supplementary file.

  1. The authors added exactly the seven more references I suggested last time. I wish the authors had checked if they are relative and searched/added more. 

Response:

All references are thoroughly checked and placed in the proper location.

  1. For Scheme 1, I am still confused if the authors prepared compounds 1to 5 by themselves (following the literature methods) or, as it was described in the Experimental (4.1), that 5a-e were used as received. If 5a-e were purchased, how come the reaction steps from 1 to 5, including the conditions? It should be easy to clarify this.

Response:

Corrected sir

  1. As to Figure 5, to my understanding,generally, the supplementary presents what is not included in the main body. To save space on the pages and the readers’ time, one is OK. If the authors like to repeat it, I am happy with it. 

Response:

We appreciate your helpful comments, but we need to use the figure within the main body of the manuscript.

Reviewer 3 Report

The Authors have made all the necessary corrections. The language is now definitely better than the one in the first submitted document.

My suggestion is to accept in the present form.

Author Response

Reviewer 3

Comments and Suggestions for Authors

The Authors have made all the necessary corrections. The language is now definitely better than the one in the first submitted document.

My suggestion is to accept in the present form.

Response:

We appreciate your considering our work for publication.